

# PyBootNet: a python package for bootstrapping and network construction

Shayan R. Akhavan[1] and Scott T. Kelley[1,2]

[1] Bioinformatics and Medical Informatics Program, San Diego State University, San Diego, CA, United States of America

[2] Department of Biology, San Diego State University, San Diego, CA, United States of America

## ABSTRACT

**Background**. Network analysis has emerged as a tool for investigating interactions among species in a community, interactions among genes or proteins within cells, or interactions across different types of data (*e.g.*, genes and metabolites). Two aspects of networks that are difficult to assess are the statistical robustness of the network and whether networks from two different biological systems or experimental conditions differ.

**Methods**. PyBootNet is a user-friendly Python package that integrates bootstrapping analysis and correlation network construction. The package offers functions for generating bootstrapped network metrics, statistically comparing network metrics among datasets, and visualizing bootstrapped networks. PyBootNet is designed to be accessible and efficient with minimal dependencies and straightforward input requirements. To demonstrate its functionality, we applied PyBootNet to compare correlation networks derived from study using a mouse model to investigate the impacts of Polycystic Ovary Syndrome (PCOS) on the gut microbiome. PyBootNet includes functions for data preprocessing, bootstrapping, correlation matrix calculation, network statistics computation, and network visualization.

**Results**. We show that PyBootNet generates robust bootstrapped network metrics and identifies significant differences in one or more network metrics between pairs of networks. Our analysis of the previously published PCOS gut microbiome data also showed that our network analysis uncovered patterns and treatment effects missed in the original study. PyBootNet provides a powerful and extendible Python bioinformatics solution for bootstrapping analysis and network construction that can be applied to microbes, genes, metabolites and other biological data appropriate for network correlation comparison and analysis.

## INTRODUCTION

Network analysis has emerged as a powerful statistical tool for investigating and interpreting complex interactions within microbial communities. The construction of networks employs mathematical algorithms to model and visualize the relationships between different microbial species, providing valuable insights into ecosystem structure, dynamics, and functional roles (*Kodera et al., 2022*). By applying network analysis techniques to microbial

Corresponding author
Scott T. Kelley, skelley@sdsu.edu

community data researchers can uncover interaction patterns among microorganisms (*Matchado et al., 2021*). Data science and the development of bioinformatics have revolutionized many fields, including the study of microbial communities. Understanding microbial community structure is crucial for advancing clinical research and gaining insights into the complex interactions within observed communities (*Schaffner, 1994*).

A network consists of circular nodes connected by lines, also known as edges. In microbial communities, nodes typically represent different microbial taxa, while edges indicate the strength and directionality of interactions between nodes (*Marai et al., 2019*). A common metric for establishing edge strength is the statistical correlation between nodes (*e.g.*, Spearman or Pearson) which can be visually represented *via* the length, color, or width of edge lines. The number of edges connected to a node, known as its degree, provides insights into the potential interactions between a specific microbe and others within the community, helping researchers understand the complex dynamics of microbial ecosystems (*Fieberg, Vitense & Johnson, 2020*). For example, in the human oral microbial community positive correlations between *Streptococcus* and *Veillonella* suggested a potential metabolic relationship where *Streptococcus* produces lactate that *Veillonella* consumes (*Browne, Shao & Lawley, 2022*). In contrast, negative correlations could indicate competition for similar resources to survive, such as *Bacteroides* and *Bacilli* in the oral cavity (*Faust et al., 2012*). Alternatively, strong correlations also indicate that organisms are subject to the same underlying environmental conditions, *i.e.,* identical growth conditions. Other metrics such as betweenness centrality and transitivity describe the architecture of networks and can be used to infer potential biological characteristics. For example, nodes with high betweenness centrality, which calculate the number of shortest paths passing through nodes, serve as valuable information for prioritizing nodes as potential therapeutic targets when identifying central transcription factors and post-translational proteins (*Sudhakar et al., 2022*). Additionally, high transitivity within the network indicates the existence of closely linked node clusters and the division of the network into separate subcomponents that provide insight into the stability of the community structures existing between species (*Loftus, Hassouneh & Yooseph, 2021*).

Two aspects of networks that are often ignored or difficult to assess are (1) the statistical robustness of the network, and (2) if networks from two different biological systems or experimental conditions are statistically different from one another. Indeed, many articles describe networks and network metrics without any statistical tests. One method that holds special promise in the statistical evaluation of networks is the bootstrap (*Efron, 1979*). Bootstrapping is a resampling technique that estimates the sampling distribution of a statistic and constructs confidence intervals. This method is particularly useful when the underlying distribution of the test statistic is unknown, when the dataset is self-contained (all the sample data is used to generate the statistic), and when dealing with population parameters other than the mean (*Schaffner, 1994*). Such a method appears to be ideal for estimating confidence intervals for networks. With networks, all the samples are used to create the networks and network metrics such as betweenness centrality and transitivity have unknown distributions. Bootstrap sampling distributions provide valuable information about the variability and uncertainty associated with a given statistic. This information is

used to construct confidence intervals, which offer a range of plausible values for the true population parameters based on the observed sample data. By employing bootstrapping techniques, researchers can make more robust inferences even when faced with limited sample sizes or complex underlying distributions (*Levin & Levina, 2019*).

Several packages and modules, such as BioNetComp (*Carvalho, 2021*) and SparCC in Python (*Friedman & Alm, 2012*) and bootnet (*Epskamp, Borsboom & Fried, 2018*) in R, have been developed to apply bootstrapping to the statistical analysis of correlation networks. While useful, these modules present certain accessibility issues. BioNetComp compares metrics between two different networks and produces network plots. However, it requires input data with a reference database to construct accurate network metrics. This proves challenging or impossible when dealing with counts of uncultured microbial taxa from an environmental community, or untargeted metabolomic data. Indeed, BioNetComp was designed specifically for interactomes from differentially expressed genes and is not generalizable for other types of datasets. SparCC implements a bootstrap method for verifying the strength of correlations within networks, but it does not bootstrap network metrics or statistically compare metrics between different networks. The R package bootnet performs bootstrap analysis with any given properly formatted feature table. However, its numerous dependencies have led to version control issues, requiring end-users to seek assistance running simple metrics. Moreover, bootnet focuses on examining the statistical robustness within networks and does not have features for comparing networks from different conditions or environments.

Here, we describe the development of PyBootNet, a flexible network bootstrapping package in Python that provides simple and intuitive functions for generating numerous bootstrapped network metrics, statistically comparing network metrics among two or more datasets, and generating network plots. The PyBootNet software package performs bootstrapping analysis on one or more feature count tables and calculates the mean and confidence intervals for seven different network metrics for every dataset. The package also provides statistical analyses such as box and whisker plots and a binomial test to determine whether individual network metrics between two different datasets are statistically different from one another. Finally, PyBootNet produces network visualizations indicating the highly supported edges and outputs the nodes with the greatest connectivity. We demonstrate the use of PyBootNet and its features using a published mouse microbiome study. While the test example is based on microbial community DNA sequence libraries, PyBootNet, is a generalizable network analysis tool that can be applied to investigate many types of biological datasets, including transcription profiles, gene function abundances, and metabolomics. PyBootNet is an intuitive and easy-to-install Python package that will enable researchers to test hypotheses of network robustness and architecture for any given correlation based biological data.

## MATERIALS & METHODS

Portions of this text were previously published as part of a preprint https://doi.org/10.1101/2024.08.08.607205.

## Data preparation

The data used to test the capabilities of PyBootNet came from *Ho et al. (2021)* and can be found at https://github.com/bryansho/PCOS_WGS_16S_metabolome. PyBootNet code, installation instructions, sample notebook, and test files can be found in Zenodo and GitHub (see 'Code Availability'). The software was developed to take input data into Python 3.10.11 functions designed to operate with Pandas DataFrames. The features for network visualization were represented as separate columns in the DataFrame. The *map_columns* function improves the readability of column values by transforming their names to a standardized format of 'X1', 'X2', 'X3', *etc*. The function maintains a dictionary that maps these transformed column names to their corresponding taxonomic information, enabling the retrieval of species-level taxonomy for each column. Next, the data was preprocessed by removing any unnecessary columns that contain numerical values.

## Bootstrapping

The *bootstrap_replicates* function generates bootstrap replicates of the data, with a default parameter of 100 iterations. In each iteration, the function randomly selects samples from the feature table with replacement from the original data (Fig. 1). In the case of microbial communities, a "sample" consists of a numerical vector where each element of the vector is a count estimate of a given bacteria in that sample. For example, a mouse gut sample dataset might comprise counts of all the bacterial genera identified *via* next generation sequencing for a single fecal sample. The resulting bootstrap replicates are stored as matrices in a list, which are used to calculate the correlations within each matrix.

## Correlation matrix and network statistics

Spearman correlation matrices were calculated for each bootstrap to replicate using the *correlation_matrix* function, considering only numerical values after removing the columns with metadata. The *calculate_network_statistics* function was used for each correlation matrix to compute various network statistics using a specified correlation threshold where the default parameter is 0.8 or higher for the strong correlations. This included the negative correlations that are −0.8 or lower. If the user only wants to indicate the positive correlations, the supplemental analysis has a positive function. Otherwise, the desired correlations are used to construct networks to calculate the following network statistics: number of edges, number of nodes, average degree centrality, transitivity, closeness centrality, betweenness centrality, and density. Each metric was stored in a dictionary for each replicate sample and each dictionary was appended to a list where the final output is a list of all the bootstrap replicate sample dictionaries.

## Network analysis and visualization

The *analyze_network_statistics* function compares network statistics across different projects. It takes a list of corresponding dictionaries of network statistics as input and generates comprehensive statistics of the data. The function calculates descriptive statistics, the mean and standard error using standard deviation, for each network statistic within each project. These results are stored in a Python pandas data frame, which is then saved as a CSV file by the function *build_network_graph* for easy access and further inspection.

## Original Feature Table

|    | B1  | B2  | B3  | B4  | B5  | B6  |
|----|-----|-----|-----|-----|-----|-----|
| S1 | 0   | 180 | 114 | 0   | 0   | 11  |
| S2 | 161 | 129 | 0   | 29  | 69  | 26  |
| S3 | 0   | 124 | 0   | 0   | 0   | 0   |
| S4 | 0   | 98  | 0   | 67  | 113 | 195 |
| S5 | 0   | 57  | 7   | 135 | 190 | 3   |
| S6 | 0   | 68  | 0   | 167 | 145 | 26  |

## Bootstrap Replicate 1

|    | B1 | B2  | B3  | B4 | B5  | B6  |
|----|----|-----|-----|----|-----|-----|
| S4 | 0  | 98  | 0   | 67 | 113 | 195 |
| S1 | 0  | 180 | 114 | 0  | 0   | 11  |
| S1 | 0  | 180 | 114 | 0  | 0   | 11  |
| S3 | 0  | 124 | 0   | 0  | 0   | 0   |
| S4 | 0  | 98  | 0   | 67 | 113 | 195 |
| S4 | 0  | 98  | 0   | 67 | 113 | 195 |

## Bootstrap Replicate 2

|    | B1  | B2  | B3  | B4 | B5  | B6  |
|----|-----|-----|-----|----|-----|-----|
| S1 | 0   | 180 | 114 | 0  | 0   | 11  |
| S2 | 161 | 129 | 0   | 29 | 69  | 26  |
| S2 | 161 | 129 | 0   | 29 | 69  | 26  |
| S4 | 0   | 98  | 0   | 67 | 113 | 195 |
| S4 | 0   | 98  | 0   | 67 | 113 | 195 |
| S1 | 0   | 180 | 114 | 0  | 0   | 11  |

**Figure 1 Example of bootstrapping an input feature table data of microbial count data.** The Original Feature Table at the top includes six samples (S1–S6) and counts of six bacterial taxa (B1–B6). The two bootstrap replicates at the bottom are the same size as the original data ($6 \times 6$) but the samples have been resampled with replacement. Any numerical feature table can be used with PyBootNet. For the test data we used the clr-transformed versions of the feature tables.

Additionally, the function creates box and whisker plots for each network statistic, allowing for visual comparison of the values across different projects. These plots are automatically saved as SVG extension image files and displayed for immediate interpretation. By providing a concise summary table and intuitive visualizations, the *analyze_network_statistics* function facilitates the exploration and understanding of network statistics across multiple projects, enabling researchers to draw meaningful conclusions and make data-driven decisions.

The *build_network_graph* function constructs a network graph based on correlation matrices, providing a visual representation of the relationships between features. It accepts either a single correlation matrix or a list of correlation matrices as input. If a list is provided, the function averages the matrices to obtain a single consolidated matrix. The function then creates a network graph using the NetworkX library (*Hagberg, Schult & Swart, 2008*), where each variable is represented as a node, and the correlations between variables are represented as weighted edges. The magnitude and sign of the correlation values determine the strength and direction of the correlations. The function allows for the filtering of significant correlations based on a specified threshold value. The resulting network graph is visualized using Matplotlib (*Hunter, 2007*), with nodes colored in sky blue and edges colored in red for negative correlations and blue for positive correlations. The edge thickness is proportional to the absolute value of the correlation. A legend is created to differentiate between positive and negative correlations. The graph is saved as an SVG file and displayed for visual inspection. This function provides researchers with a powerful tool to explore
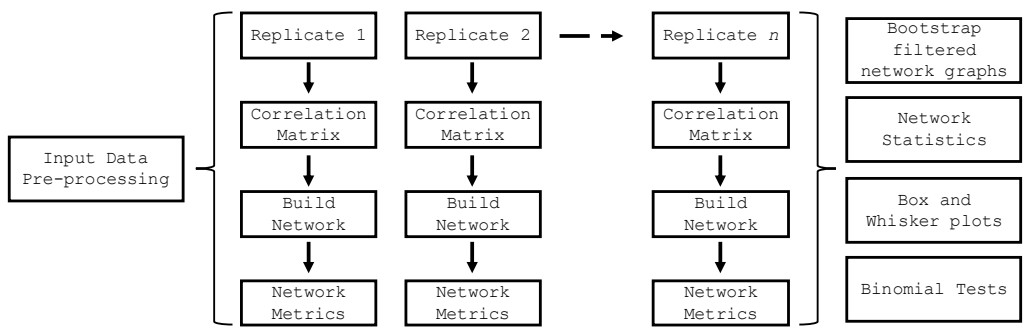

**Figure 2** **PyBootNet workflow of creating replicates and calculating correlations and network metrics (stats).** Network construction and stats are generated from combining the individual bootstrap results.

**Table 1** **Description of *PyBootNet* functions and analyses.** The names of the functions are paired with the description of their use including input and output.

| Functions | Description |
|---|---|
| map_columns() | Changes the values in a specified column of a DataFrame to be more readable, mapping them to 'X1', 'X2', 'X3', *etc.* |
| bootstrap_replicates() | Creates bootstrap replicates of the input data. The default number of iterations is 100. |
| correlation_matrix() | Calculates the Spearman correlation matrix for each DataFrame in the input list, taking in only numerical values. |
| calculate_network_statistics() | Calculates network statistics for each correlation matrix in the input list, using a specified threshold. |
| analyze_network_statistics() | Analyzes the network statistics for different projects, calculates descriptive statistics, and creates box plots for each statistic. |
| build_network_graph() | Builds a network graph from the input correlation matrices, averaging them if multiple matrices are provided. |
| top_nodes() | Identifies the top nodes with the highest degree in the network graph. |
| net_stat_binomial_test() | Performs a binomial test on the network statistics of two sets of bootstrap replicates. Takes the outputs of the calculate_network_statistics. |

and understand the complex relationships between variables in their data, facilitating the identification of key connections and potential patterns. The *top_nodes* function identified the nodes with the highest degree in the network graph, while the *most_connected_nodes* function determined the most connected node. The *nodes_edges_table* function created a table of the number of edges for each node. The PyBootNet workflow is illustrated in Fig. 2 and descriptions of the primary functions can be found in Table 1.

## Data export and statistical testing

The *save_table_to_csv* function was implemented to store preprocessed data as a CSV file for import. A binomial test was performed on the network statistics of two sets of bootstrap replicates using the *net_stat_binomial_test* function, which takes the outputs

of the *calculate_network_statistics* function. The binomial test is conducted using a list, where each element of the list is a dictionary containing the calculated network statistics. Specifically, we use a two tailed binomial test to perform pairwise comparisons for every network statistic. For example, we can test the hypothesis that two different networks differ in their total number of edges, with the null hypothesis being no difference. We test this hypothesis by calculating the number of edges for each bootstrap replicate for the two networks, then for each bootstrap replicate we ask if the number of edges in network 1 is greater than network 2 or vice versa. If there is no difference, we would expect over many bootstrap replicates that the number of times network 1 would have more edges than network 2 would be the approximately the same as the number of times network 2 has more edges than network 1. However, if we run 500 bootstrap replicates and the number of edges in network 1 is always greater than network 2, this is equivalent to flipping a coin 500 times and always coming up tails and the binomial test is designed for determining the probability of such binary outcomes.

### Additional analyses

The *build_positive_network* function constructs a network graph focusing on positive correlations above a specified threshold, while the *build_filtered_networks* function builds a filtered network graph, retaining only nodes with a maximum degree specified by the parameter *max_degree*. The *build_negative_networks* function creates a network graph focusing on negative correlations below a specified threshold, and the *negative_filtered_networks* function builds a filtered network graph of negative correlations, where both parameters have the same values. The b*ootstrap_sample_with_correlation* function performs bootstrapping on the input data, calculates correlations for each bootstrap replicate, and returns the average correlation matrix. Finally, the *top_nodes_network_graph* function constructed a network graph highlighting the top nodes with the highest degree of centrality. The default parameter is set to 20 nodes, but the user can adjust accordingly. All functions were designed to provide flexibility in input data format and output file formats, facilitating integration into the analysis pipeline. Input parameters and file names were adjusted according to the specific use case of the thesis.

### Computational testing platform

PyBootNet analysis was run on Windows 10 Home 64-bit operating system with an AMD Ryzen 7 5800X 8-core processor at 3.8 GHz clock speed. The dedicated memory available was 32 GB and the graphical processor was an Nvidia GeForce RTX 3080 with 10 GB of video memory.

## RESULTS AND DISCUSSION

To demonstrate the utility of PyBootNet, we performed bootstrap network analyses on bacterial community data obtained from the *Ho et al. (2021)* mouse gut microbiome study. Ho et al. used a hyperandrogenic female mouse model to study how high levels of androgens (hyperandrogenism) impacted a gut microbiome. Hyperandrogenism is a key feature of polycystic ovary syndrome (PCOS), a reproductive and metabolic disorder experienced

by ~10% of reproductive-aged women worldwide (*Lizneva et al., 2016*). In addition to ovarian cysts, women with PCOS often have oligomenorrhea or amenorrhea and are a risk for metabolic disorders (*The Rotterdam ESHRE/ASRM-Sponsored PCOS Consensus Workshop Group, 2004*). The authors used the drug letrozole, implanted surgically under the skin to release a constant dose for the entirety of the study, to induce hyperandrogenism. Letrozole inhibits the conversion of estrogen to androgens (*e.g.*, testosterone) resulting in high levels of endogenous androgens in female mice and, subsequently, the hallmarks of PCOS including oligo- or anovulation, polycystic ovaries, elevated luteinizing hormone (LH) levels, weight gain, abdominal adiposity, and hyperinsulinemia (*Kauffman et al., 2015*).

Previous studies demonstrated significant differences in the gut microbiome in both the letrozole model and in human women with PCOS compared to placebo controls and non-PCOS healthy controls (*Torres et al., 2018*). Another paper also showed strong evidence that altering the microbiome through cohousing could alleviate the symptoms of PCOS in the mice even in the face hyperandrogenemia (*Torres et al., 2019*), indicating that the microbiome may play an important role in aspects of the PCOS phenotype. In the cohousing experiment, letrozole and placebo mice were placed in the same cages and, because mice are coprophagic, the gut microbiome was shared between the animals. Torres et al. showed that the letrozole mice that were co-housed with the placebo mice significantly improved or eliminated the PCOS-like reproductive and metabolic symptoms in the face of continuous letrozole treatment.

*Ho et al. (2021)* performed a multiomics analysis of the fecal microbiome samples collected from the cohousing study. This included an integrated analysis of the microbiomes and metabolomes of fecal samples from the four treatment groups collected during puberty and afterwards: Letrozole (LET), Placebo (PLA), cohoused-LET (co-L), and cohoused-PLA (co-P). The study confirmed that the taxonomic compositions of the microbiomes and the molecular composition of the metabolomes of the LET and PLA samples differentiated strongly during puberty (time 2; 6-weeks-old). However, it also showed that the co-L and co-P microbiomes and metabolomes converged during puberty and that the pattern was stronger with the metabolome than the microbiome. Interestingly, this pattern proved temporary, and post-puberty (experiment time 5; 9-weeks-old) there was much less differentiation in the metabolome and no detectable differentiation in the microbiome.

Here, we applied PyBootNet to analyze microbiome network stability based on the bacterial 16S ribosomal RNA based taxonomic datasets within each treatment group at the two different time points, and to statistically compare networks metrics within and between timepoints. All the following results were generated using 500 bootstrap replicates with a minimum correlation value of $r = 0.8$. Overall, the bootstrapped 16S bacterial network analyses revealed may novel and surprising patterns that could not be inferred from other types of statistical analyses. For example, while LET and PLA samples at time 2 were very different in terms of taxonomic composition, they had very similar network structures (Fig. 3). On the other hand, while co-L and co-P taxonomic composition converged, there were stark differences in the networks (Fig. 4). Interestingly, the co-L network revealed interesting clusters of strongly negative correlations (Fig. 4A), while virtually no strong

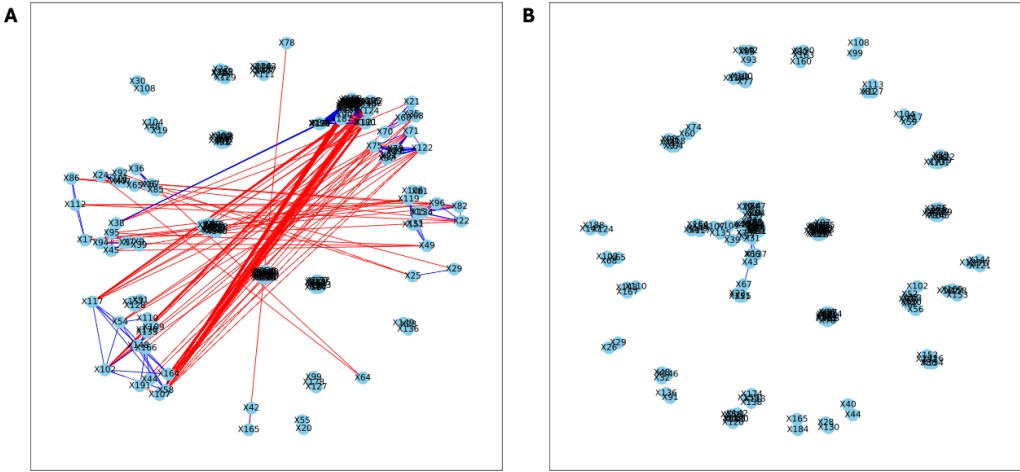

**Figure 3  Bootstrapped correlation networks of fecal microbial communities.** Letrozole (A) and Placebo (B) network visualizations for fecal samples collected during puberty (time 2). The nodes are bacterial species, and the edges are positive (blue) and negative (red) correlations.

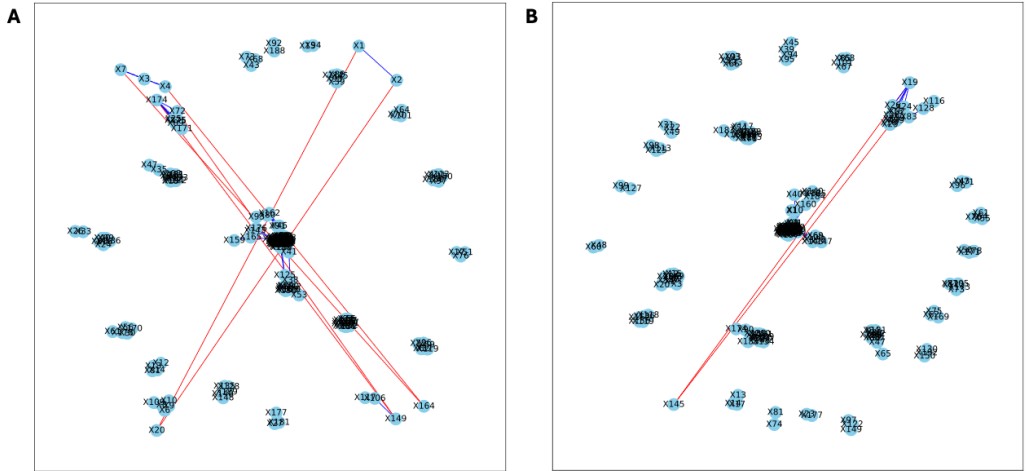

**Figure 4  Bootstrapped correlation networks of fecal microbial communities.** Co-housed Letrozole (A) and co-housed Placebo (B) network visualizations for fecal samples collected during puberty (time 2). The nodes are bacterial species, and the edges are positive (blue) and negative (red) correlations.

negative correlations were observed in the time 2 co-P network (Fig. 4B) or any of the other networks. The network metrics were also revealing. The LET and co-L networks had more edges, higher density, and greater degree centrality than either the PLA or co-P networks, indicating that the effect of hyperandrogenism on microbial correlations was not disrupted by cohousing coprophagy (Fig. 5). Table 2 shows the results of binomial tests for the pairwise comparisons of the time 2 puberty network metrics.

The analysis of the post-puberty (time 5) networks also provided novel insights. All the networks were more complex with many more correlations, both positive and negative

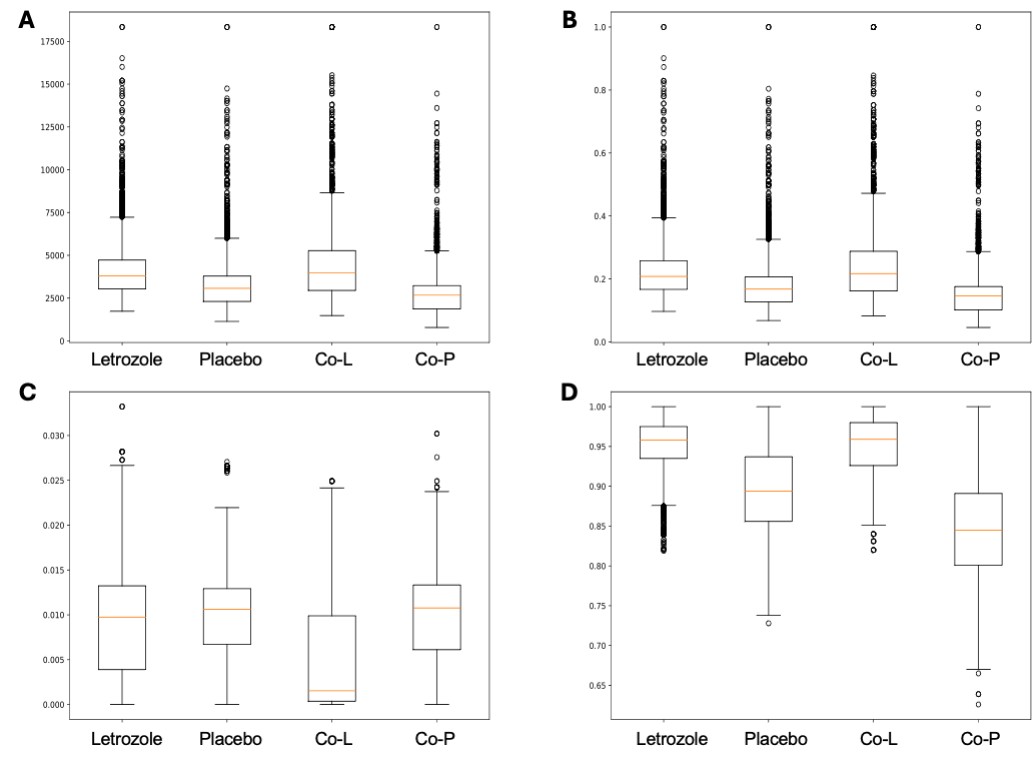

**Figure 5** Box and whisker plots for bootstrapped network statistics for samples collected during puberty (time 2). The plots compare number of edges (A), average degree centrality (B), betweenness centrality (C), and transitivity (D).

**Table 2 Binomial tests results comparing mouse fecal microbial networks.** Binomial bootstrap tests results ($n = 500$ replicates) of pairwise comparisons of *Ho et al. (2021)* treatment groups at time 2 (puberty).

| Network metric | co-L *vs.* co-P | | LET *vs.* PLA | | co-L *vs.* LET | | co-P *vs.* P | |
|---|---|---|---|---|---|---|---|---|
| | Statistic | *p*-value | Statistic | *p*-value | Statistic | *p*-value | Statistic | *p*-value |
| Number of edges | 0.759 | 3.9 e-308 | 0.674 | 1.2 e-136 | 0.104 | 2.2 e-5 | 0.634 | 8.8 3-69 |
| Degree centrality | 0.758 | 3.7 e-305 | 0.673 | 5.1 e-136 | 0.475 | 4.8 e-5 | 0.623 | 1.5 e-68 |
| Betweeness centrality | 0.302 | 4.1 e-177 | 0.504 | N.S. | 0.673 | 2.1 e-135 | 0.486 | N.S. |
| Transitivity | 0.905 | 0.0* | 0.808 | 0.0* | 0.482 | 1.3 3-2 | 0.711 | 1.2 e-201 |

**Notes.**
*\*p*-value below e-320.
N.S., not significant.

(Figs. 6 and 7). However, the PLA network metrics were consistently lower than all the other groups, apart from the betweenness centrality, while the co-P network metrics were like the LET and co-L sample networks (Fig. 8). The clear difference in the PLA network compared to all the others suggests an important role of the estrous cycle in structuring the microbial communities. A recent paper by *Sisk-Hackworth et al. (2024)* that also used PyBootNet identified a similar pattern when comparing the gut microbial communities of wild-type females to mutant females that had their reproductive axis ablated as well as

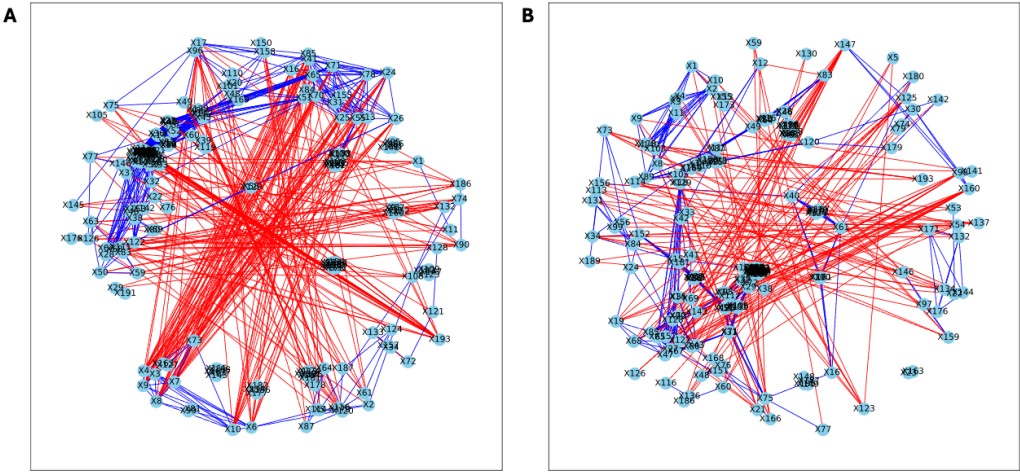

**Figure 6** **Bootstrapped correlation networks of fecal microbial communities.** Letrozole (A) and Placebo (B) network visualizations for fecal samples collected post-puberty (time 5). The nodes are bacterial species, and the edges are positive (blue) and negative (red) correlations.

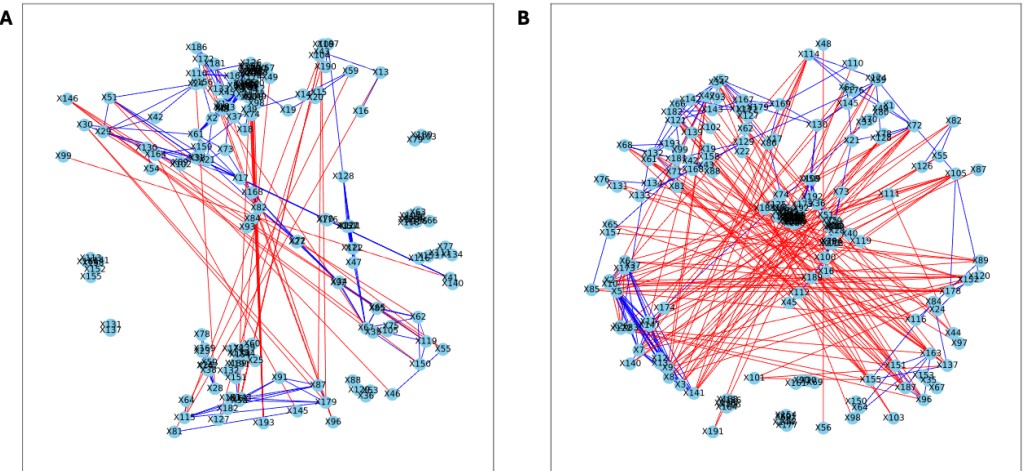

**Figure 7** **Bootstrapped correlation networks of fecal microbial communities.** Co-housed Letrozole (A) and co-housed Placebo (B) network visualizations for fecal samples collected post-puberty (time 5). The nodes are bacterial species, and the edges are positive (blue) and negative (red) correlations.

and to males, neither of which have an estrous cycle. These two results suggest that the estrous cycle introduces a higher level of variability in the microbial community which would reduce the overall network complexity (*e.g.*, number of edges, degree centrality). While the co-P female mice also had an estrous cycle, they were also ingesting microbes from their co-L cage mates which may have stabilized the network complexity.

Overall, PyBootNet identified striking differences in architectural network not perceived with prior statistical analyses. Biologically, these results suggest novel interesting effects of hyperandrogenism and co-housing on interactions, potentially strong patterns of

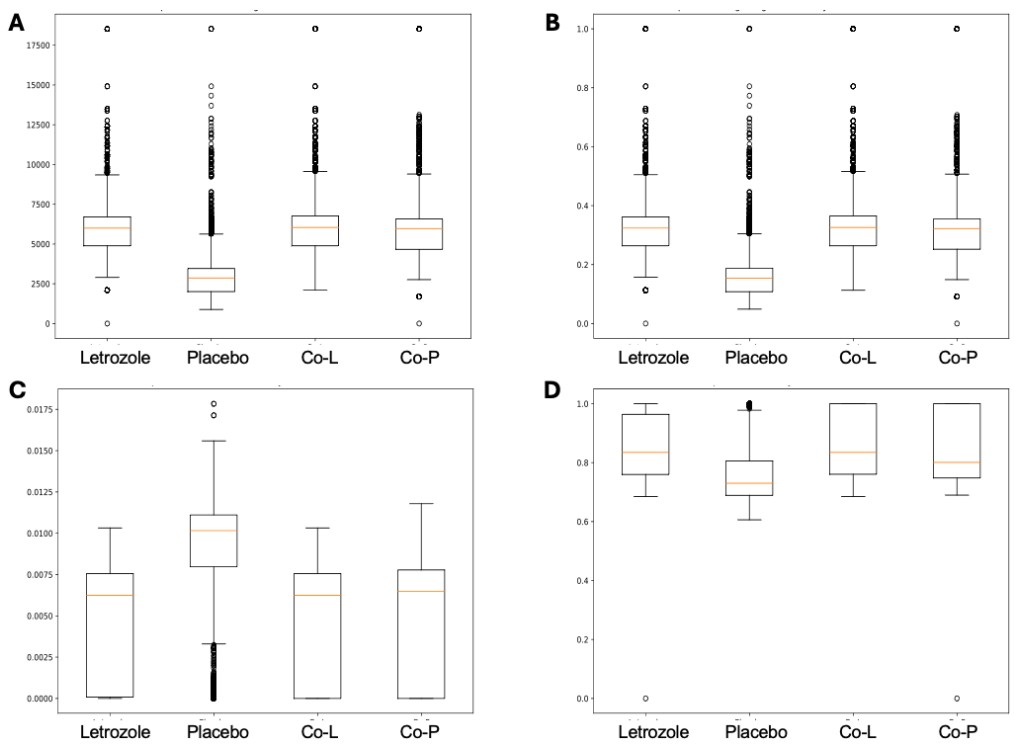

**Figure 8  Box and whisker plots for bootstrapped network statistics for samples collected post-puberty (time 5).** The plots compare number of edges (A), average degree centrality (B), betweenness centrality (C), and transitivity (D).

competition between groups of bacteria (*i.e.,* the high degree of negative correlations), and differences in network metrics worthy of further exploration.

## CONCLUSIONS

PyBootNet provides a much needed, user-friendly, and efficient Python package for network bootstrap analysis and construction. The intuitive application of PyBootNet to the published mouse gut microbiome study demonstrates the package's ability to determine interesting new biological insights and demonstrates its potential for wider use. The software's ability to generate bootstrapped network metrics, statistically compare network metrics among datasets, and visualize networks enables researchers to formulate data-driven hypotheses and uncover meaningful patterns in microbial ecosystems and statistically compare correlation networks for any type of data. PyBootNet's minimal dependencies, clear function design, and straightforward input requirements further contribute to its accessibility and ease of use. The development of PyBootNet addresses the limitations of existing packages, such as BioNetComp and BootNet, by offering a more flexible and generalizable Python-based solution for bootstrapping analysis and network construction. By integrating these functionalities into a single, user-friendly package, PyBootNet streamlines the analysis process and facilitates the exploration of

microbial community interactions. Also, because this code is written in Python and all the bootstrapped networks and metrics are analyzed independently, PyBootNet will be easy to parallelize in future versions. As PyBootNet continues to be refined and expanded, it has the potential to become an essential tool for network analysis of many biological systems.

## ACKNOWLEDGEMENTS

We thank L. Sisk-Hackworth and K. Nannini for providing the data used to test and enhance PyBootNet's capabilities. L. Miller provided essential guidance for calculating standard errors, C. Zúñiga guided network construction, and both provided helpful suggestions on the text. Members of the Kelley Lab offered valuable insights on developing new software and helping the lead author stay on track.

### Funding

The authors received no funding for this work.

### Competing Interests

The authors declare there are no competing interests.

### Author Contributions

- Shayan R. Akhavan performed the experiments, analyzed the data, prepared figures and/or tables, authored or reviewed drafts of the article, and approved the final draft.
- Scott T. Kelley conceived and designed the experiments, prepared figures and/or tables, authored or reviewed drafts of the article, and approved the final draft.

### Data Availability

The code and data is available in GitHub and Zenodo:

– https://github.com/Shayan-Akhavan/PyBootNet.git

– Akhavan, S., & Kelley, S. T. (2024). PyBootNet: A python package for bootstrapping and network construction. Zenodo. https://doi.org/10.5281/zenodo.14247208.

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
