# Peer review of "PyBootNet: a python package for bootstrapping and network construction"

_PeerJ, doi:10.7717/peerj.18915_

## Round 0.1 · original submission · Major Revisions

Before a decision can be made regarding the acceptance of your manuscript for publication in PeerJ, major revisions are required in response to the reviewer comments. Reviewer 1 provides a separate file that details their concerns and provides suggestions for improvement of your manuscript. While a change in the focus of the manuscript away from simply describing the PyBootNet package as indicated by reviewer 2 is not required, it would be useful to provide a more detailed example of the application of PyBootNet to a real-world dataset, describing its use, results obtained, and how those results and the conclusions derived from those results were enhanced by the use of PyBootNet. It is also important to ensure that all example datasets provided through GitHub are complete, thoroughly documented, and able to accurately demonstrate the functionality of the package. Finally, we have also received a suggestion that the package name, PyBootNet is too close to, and may be confused with, the R package “bootnet”. The reason for the suggestion is that the established R package bootnet was described as having limited functional overlap with PyBootNet, and the similarities in the name could lead to confusion by potential users.

Reviewer 1 ·

Basic reporting

The manuscript is generally well-written in clear and unambiguous English, making it easy to understand. However, there are some issues that need to be addressed. Some literature references are missing, and additional references that provide biological context and discussion of the results are needed. The figures should be reformulated as some do not add value to the main manuscript. Additionally, the results need to be better described and compared to the hypotheses.

Experimental design

The package design is well established and provides new ways of analysis.But the package manual must be redone and better clarified. The current mode is confusing and there is a lack of analysis notebooks.

Validity of the findings

The impact and how the package aggregated new results is not well explained in the text.

Annotated reviews are not available for download in order to protect the identity of reviewers who chose to remain anonymous.

·

Basic reporting

No comment - the article is clear and well structured.

Experimental design

The article is more of a description of how a specific Python library can be used than a research paper. The question appears to be can the Python package PyBootNet do what it claims to do. As a test, which is more of a demonstration, microbiome data are analyzed with the software. As expected , the software presents results. These results are presented in tables, but there is no discussion of what the results indicate other than the software can produce results.

Therefore the article fails for not being within the Aims and Scope of the journal, and fails for not being a well defined, relevant, and meaningful research question.

I suggest the focus of the paper be changed to a research question on microbiomes where the Python package is the tool to analyze the research data and not the focus of the "research."

Validity of the findings

The Python package PyBootNet is used to analyze microbiome data, but I am not sure what values are analyzed. The paper states using the statistical correlation between nodes. What numeric data represents the nodes? What numerical metrics are used to run the statistical correlations? I suggest providing a better description of the data since "mouse gut microbiome data" can be represented by many numerical metrics.

The conclusion supports that there is no real research question other than demonstrating that this Python software package will compute what it claims to compute. I argue that although convenient, there are other software packages that can be used to analyze microbiome data. Of course I still need to know what specific data we are analyzing.

Since there is no real research question and the results are just that the software produces the output that it claims to produce, the paper fails for validity of findings. Again I suggest the focus of the paper be changed to a research question on microbiomes where the Python package is the tool to analyze the research data and not the focus of the "research."

Additional comments

Although PyBootNet provides functions to create bootstrap data, there are also functions in R that will do the same. However that point is irrelevant since I argue the paper should focus on a research question and not the tool that was used to analyze the data.

Reviewer 3 ·

Basic reporting

No comments

Experimental design

No comments

Validity of the findings

No comments

Additional comments

No comments

---

## Round 0.2 · Minor Revisions

Before your manuscript can be accepted for publication, a response to the critique of Reviewer 1 is required along with a revised manuscript that addresses the reviewer's concerns. Significant changes are not needed, and any necessary manuscript revisions should not require extensive updates. You should consider the reviewer's suggestions as areas in which the manuscript can be made a bit more clear.

Reviewer 1 ·

Basic reporting

1. The authors made significant modifications to the project, but still need to better explain the biological results, comparisons between packages and better prepare the presentation of the results.

2. The example notebook must be further reformulated. You can see that the main notebook does not have text jupyter notes that explain the line comands (https://github.com/Shayan-Akhavan/pybootnet/blob/main/gut_analysis_notebook.ipynb).

You should use the good practice of creating a software manual. You must divide the notebook into sections, and in each of them place notes explaining the functionality of the code below, and exemplifying the findings, leaving them consistent. As it stands, it's just code and more code.

Additionally, the USAGE in main must refer to the notebook in question.

Experimental design

1. The authors did not respond by comparing the features with the bootnet package in R. Both have similar features, even for metric calculations. Furthermore, the Bionetcomp package was designed in Python and not in R as they said. I suggest adding a table comparing all packages that perform similar features, the correct language used, and the differences. It is essential for comparisons between software. An article with new features that does not compare its features with other packages is unpublishable.

They didn't answer this question correctly in reviewer 1's answer: "Finally, we have also received a suggestion that the package name, PyBootNet is too close to, and may be confused with, the R package “bootnet”. The reason for the suggestion is that the established R package bootnet was described as having limited functional overlap with PyBootNet, and the similarities in the name could lead to confusion by potential users".

2. The paragraph that describes possible new findings (line 279 to 300) does not have concrete biological explanations, only descriptive explanations about the behavior of the network, metrics, and no biological applications. Authors must actually identify novelties using the biological relationships between co-P, co-L, PLA and LET.

3. The previous paragraph (line 242 to 277) is confusing. It is not demonstrated how PyBootNet helped in the biological explanation of the findings of Torres, 2018. The authors must show biological findings together with the network composition findings.

4. The authors say they bring novelties in line 296 to 300 but it is not yet clear which ones. I still have difficulty observing biological relevance when using PyBootNet.

5. The authors placed the network nodes with an 'X' prefix coming from the use of the pandas library in Python, which places an X in numeric columns. They must modify so that care is taken when handling the dataset for publication. Use the word 'condition' os the letter 'C' and indicate it in the figure legend.

Validity of the findings

I would be happier with creating another application test case.

---

## Round 0.3 · accepted · Accept

Thank-you for your timely response to the critiques of your last revision. After review, I have determined that you have adequately addressed all of the reviewer comments and I am happy with the current version. The manuscript is now ready for publication.